# A Novel Method of Trajectory Optimization for Underwater Gliders Based on Dynamic Identification

**Ming Yang** [1,2], **Yanhui Wang** [1,2,*], **Yan Liang** [1], **Yu Song** [2] and **Shaoqiong Yang** [1,2]

1   Key Laboratory of Mechanism Theory and Equipment Design of Ministry of Education, School of Mechanical Engineering, Tianjin University, Tianjin 300350, China; mingyang@tju.edu.cn (M.Y.); liangyan312@tju.edu.cn (Y.L.); shaoqiongy@tju.edu.cn (S.Y.)
2   The Joint Laboratory of Ocean Observing and Detection, Pilot National Laboratory for Marine Science and Technology, Qingdao 266237, China; ysong2@qnlm.ac
*   Correspondence: yanhuiwang@tju.edu.cn

**Abstract:** For underwater gliders (UGs), high trajectory accuracy is an important factor in improving the observation of ocean phenomena. In this paper, a novel method of trajectory optimization is proposed to increase the trajectory accuracy of UGs, which is approximately based on the nonlinear dynamic model, rather than the linearization model. Firstly, a dynamic model of UGs is established to analyze the effect of the input parameters on the trajectory error, based on some approximate models that replaced the dynamic model due to its strong nonlinearity. Then, an identification strategy for the trajectory error is proposed, and the trajectory optimization strategy is analyzed while considering gliding range loss and observation distance loss. Finally, the identification strategy and trajectory optimization strategy proposed in this paper are verified by a sea trial of Petrel-L. The dynamic model, identification strategy, and optimization strategy are appropriate for other UGs.

**Keywords:** dynamic model; underwater glider; dynamic identification; trajectory error

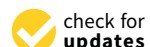



## 1. Introduction

With the development of marine science and technology, the scientific expedition vessel is no longer the only way of exploring the ocean. In recent years, autonomous underwater exploration has become a worldwide research hotspot. Therefore, various types of underwater equipment have been developed, such as autonomous underwater gliders, remotely operated vehicles [1], and manned submersibles [2]. The underwater glider (UG) [3] is one of the most promising ocean observation platforms, and it has been widely applied to long-duration and long-range ocean phenomena. A UG can ascend and descend in the water column, realizing horizontal motion by controlling its attitude so that its wings always generate lift. Thus, its motion is a zigzag. Since the concept of the UG was first proposed in 1989, various types of UGs have been developed successfully, such as the Slocum [4], Seaglider [5], Spray [6], Sea-Explorer [7] and Petrel-L [8].

In practical applications, UGs are required to follow planned trajectories, which are determined by specific mission requirements. Thus, the trajectory-keeping ability is an important index for UGs to better serve the observation of ocean phenomena. Generally, dynamic modeling provides an efficient way to study the trajectory accuracy of UGs, which has been established successfully with many methods, such as the Newton–Euler method [9], second Lagrange equation [10], Gibbs–Appell equation [11], differential geometry method [12], and Kirchhoff equation [13].

Based on the dynamic model, much significant research on the trajectory analysis of UGs has been carried out. Wu et al. [14] established a dynamic model of UGs validated by experimental data and studied an analysis method and a compensation strategy of glider motion accuracy. Leonard and Graver [9] derived a dynamic model of UGs to study stability and controllability of glide paths and to derive feedback control laws based on

linearization. Ziaeefard [15] presented a novel roll mechanism and an efficient control strategy for UGs using multiple feedforward-feedback controllers. Mahmoudian [16] presented an approximate analytical expression for steady-turning motion for a realistic UG model. Mahmoudian and Woolsey [17] described the dynamic modeling of the UG and the numerical implementation of a motion control system with approximate analytical expressions for wings-level and turning flight. Sang et al. [18] presented a new hybrid heading tracking control algorithm, which integrated an adaptive fuzzy incremental PID and an anti-windup compensator to improve the adaptability and robustness of an underwater glider's heading control. Lyu et al. [19] established a dynamic model by considering the buoyancy and pitch-regulating system and investigated the impact of the winglet on hydrodynamic performance and gliding trajectory of a blended-wing-body UG. Wu et al. [20] established a dynamic model and studied a multi-objective optimization method to determine the control parameter values that improve the performance of the glider. Smith et al. [21] investigated the implementation of a large-scale, regional ocean model into the trajectory design for autonomous gliders to improve their navigational accuracy. Wang et al. [22] designed a linear-quadratic regulator for the underwater glider and studied the motion path of the feedback control system in an ocean vertical plane. De la Cruz and Torres [23] presented a pitch-based depth-tracking controller for a hybrid UG propelled by a constant forward force applied through a single thruster. Yoon and Kim [24] addressed the optimization of the longitudinal trajectory of a UG operating in water of limited depth, obtaining the minimum-time trajectory to achieve the maximum-advance speed.

Although the above research reported some results about the trajectory of UGs, few works consider the following problems, which are summarized from the sea trial data of the Petrel-L, a UG developed by Tianjin University, China. Generally, UGs only carry an electronic compass for underwater navigation, which can measure the attitudes of UGs, including the pitch angle, roll angle, and heading. Disturbed by the ocean environment or calibration error, the heading of a UG will frequently change, and the UG will adjust to the planned value when the difference value is large enough, which will lead to the deviation from the planned trajectory. The UG cannot return to the planned trajectory on the premise that it is lacking an underwater positioning device. Thus, the problems to be solved can be summarized as: (1) What is the relationship between the trajectory of the UG and the sequence of heading adjustments? (2) How can we compute the deviation of the vehicle trajectory from the planned trajectory? (3) What motion strategies can the UG adopt to return to the planned trajectory?

To fill this research gap, this paper proposes a novel method to decrease the deviation of UGs from the planned trajectory and improve the UGs' trajectory accuracy. First, a dynamic model is established to analyze the effect of the heading adjustment on the trajectory deviation. Then, an identification strategy is presented to obtain the distance of the UG to the planned trajectory with the surrogate model method. Next, a trajectory optimization is carried out to study the advantages and disadvantages of a range of heading adjustments. Finally, a sea trial of Petrel-L verifies the correctness of the method proposed in this paper. The main contributions of this paper are summarized as follows.

(1) The effect of the heading adjustment, realized by rotating the internal mass block of the Petrel-L UG, on its trajectory error is studied with the dynamic model.
(2) A systems identification is presented for the first time to obtain the deviation distance of UGs caused by heading adjustment to the planned trajectory.
(3) To improve the trajectory accuracy of UGs, trajectory optimization schemes are contrasted while considering gliding range loss and observation distance loss.
(4) The proposed method is verified using data from a Petrel-L sea trial.

The rest of this paper is organized as follows. Section 2 establishes the dynamic model of UGs. In Section 3, the identification strategy is introduced. Section 4 gives the trajectory optimization results and discussion, and Section 5 describes the test verification. Section 6 concludes the paper with a discussion of the future work.

## 2. Dynamic Modeling of Petrel-L

As shown in Figure 1, this paper takes the Petrel-L [8,25,26], a long-range UG developed by Tianjin University, China, as the study object. It consists of a cylindrical pressure vessel, two fixed wings, two fairings, a vertical stabilizer, and an antenna. After some optimization of hydrodynamic shape [27], pressure hull design [28], motion parameter [8,29] and multidisciplinary design [30], the gliding range of Petrel-L reached over 4000 km.

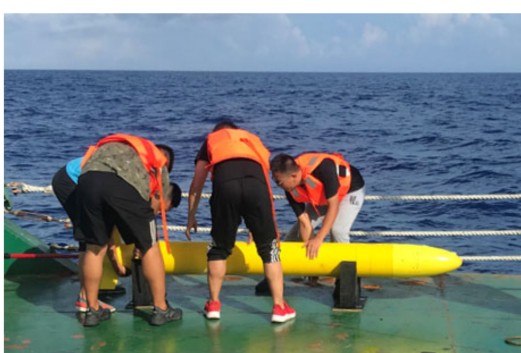

**Figure 1.** Petrel-L glider in South China Sea.

To analyze the trajectory accuracy of Petrel-L, a dynamic model needs to be established. First, the coordinate frames of Petrel-L are defined to facilitate the subsequent deduction, as shown in Figure 2. Three frames, including the inertial frame, *E-XYZ*, body frame, *O-xyz*, and velocity frame, *O'-x'y'z'*, are defined. The directions of all the coordinate axes in one frame are shown in Figure 2, which obey the right-hand rule.

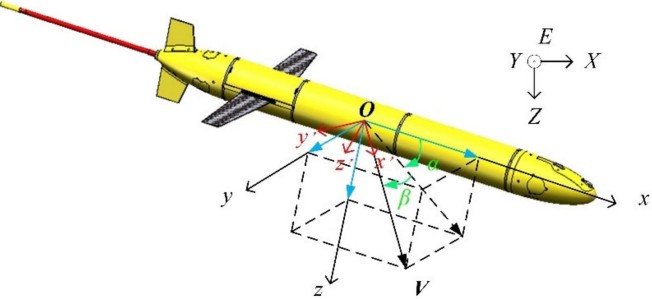

**Figure 2.** Coordinate frames of Petrel-L.

### 2.1. Kinematics

In the inertial frame, the position and the attitude of Petrel-L are expressed by the position vector, $\boldsymbol{b} = [X, Y, Z]^{\mathrm{T}}$, and the angle vector, $\boldsymbol{\eta} = [\varphi, \theta, \psi]^{\mathrm{T}}$, where the roll angle, $\varphi$, pitch angle, $\theta$, and heading, $\psi$, denote the angles that the body frame rotates around the *X*-axis, *Y*-axis, and *Z*-axis, respectively, from the attitude, coinciding with the internal frame. The inertial frame, *E-XYZ*, can be coincident with the body frame, *O-xyz*, after rotating three times and following the rotation matrix of [31].

$$\boldsymbol{R}_{\mathrm{B}}^{\mathrm{E}} = \begin{bmatrix} \cos\psi\cos\theta & \sin\psi\cos\theta & -\sin\theta \\ \cos\psi\sin\theta\sin\varphi - \sin\psi\cos\varphi & \sin\psi\sin\theta\sin\varphi + \cos\psi\cos\varphi & \cos\theta\sin\varphi \\ \cos\psi\sin\theta\cos\varphi + \sin\psi\sin\varphi & \sin\psi\sin\theta\cos\varphi - \cos\psi\sin\varphi & \cos\theta\cos\varphi \end{bmatrix} \quad (1)$$

Thus, the rotation matrix from the body frame to the inertial frame is the transpose of $R_{\text{B}}^{\text{E}}$. The velocity vector of Petrel-L in the inertial frame can be expressed as

$$
\begin{aligned}
X' &= u\cos\psi\cos\theta + v(\cos\psi\sin\theta\sin\varphi - \sin\psi\cos\varphi) \\
&\quad + w(\cos\psi\sin\theta\sin\varphi + \sin\psi\sin\varphi) \\
Y' &= u\sin\psi\cos\theta + v(\sin\psi\sin\theta\sin\varphi + \cos\psi\cos\varphi) \\
&\quad + w(\sin\psi\sin\theta\cos\varphi - \cos\psi\sin\varphi) \\
Z' &= -u\sin\theta + v\cos\theta\sin\varphi + w\cos\theta\cos\varphi
\end{aligned}
\tag{2}
$$

The motion trajectory, $X(t)$, $Y(t)$, and $Z(t)$, of Petrel-L in the inertial frame can be expressed by integrating Equation (2).

In the body frame, the velocity and the angular velocity of Petrel-L can be expressed by the velocity vector, $V = [u, v, w]^{\text{T}}$, and the angular velocity vector, $\Omega = [p, q, r]^{\text{T}}$, involving velocities and angular velocities along the $x$-axis, $y$-axis, and $z$-axis, respectively. The relationship between body frame and velocity frame is determined by the angle of attack, $\alpha$, and the sideslip angle, $\beta$, shown in Figure 2, and the rotation matrixes between, which can be expressed as

$$
R_{\text{V}}^{\text{B}} = \left( R_{\text{B}}^{\text{V}} \right)^{\text{T}} = \begin{bmatrix} \cos\alpha\cos\beta & \sin\beta & \cos\beta\sin\alpha \\ -\cos\alpha\sin\beta & \cos\beta & -\sin\alpha\sin\beta \\ -\sin\alpha & 0 & \cos\alpha \end{bmatrix}
\tag{3}
$$

Similarly, the relationship between the change rate of the attitude angles and the angular velocities can be expressed as

$$
\begin{aligned}
\varphi' &= p + q\tan\theta\sin\varphi + r\tan\theta\cos\varphi \\
\theta' &= q\cos\varphi - r\sin\varphi \\
\psi' &= (q\sin\varphi + r\cos\varphi)/\cos\theta
\end{aligned}
\tag{4}
$$

The attitude of Petrel-L in the inertial frame, $\varphi(t)$, $\theta(t)$, $\psi(t)$, can be obtained by integrating Equation (4).

The velocity, $V$, angle of attack, $\alpha$, and the sideslip angle, $\beta$, are respectively expressed as

$$
V = \sqrt{u^2 + v^2 + w^2}
\tag{5}
$$

$$
\alpha = \arctan(w/u)
\tag{6}
$$

$$
\beta = \arcsin(v/V)
\tag{7}
$$

### *2.2. Force Analysis*

When Petrel-L travels in the seawater, it is subject to external forces, including the buoyancy, weight, inertial forces, and viscous hydrodynamic forces. In addition, there exist interactive forces between the UG body and the movable internal mass block. All forces are expressed relative to a local origin, $O$. Before force analysis, an assumption is adopted that the displacement of Petrel-L is constant, and the mass of Petrel-L is variable when the buoyancy changes.

### 2.2.1. Buoyancy Device and Attitude Device

As shown in Figure 3a, the buoyancy device (BD) can transfer oil between the internal oil tank and external bladder, which results in the net buoyancy variation required by Petrel-L for dive or climb motion.

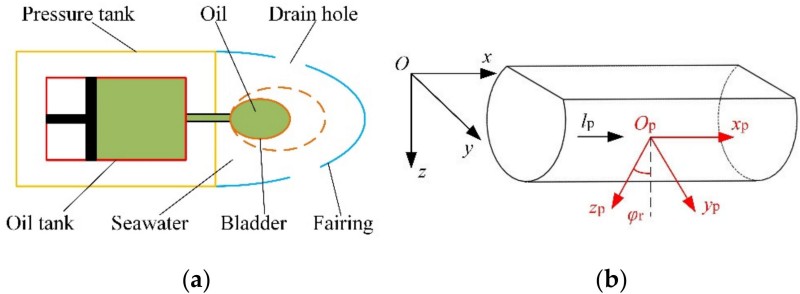

**Figure 3.** Buoyancy control device (**a**) and attitude control device (**b**).

When Petrel-L is neutrally buoyant at the sea surface, the volume of oil inside the oil tank is denoted as $V_{bN}$. Thus, when the oil volume inside the internal oil tank is $V_b$, the mass variations of the internal oil tank, $\Delta m_{bin}$, and the external bladder, $\Delta m_{bout}$, are expressed as

$$\Delta m_{bin} = (V_b - V_{bN})\rho_{oil} \tag{8}$$

$$\Delta m_{bout} = -\Delta m_{bin} + \frac{\Delta m_{bin}}{\rho_{oil}}\rho_w = (V_b - V_{bN})(\rho_w - \rho_{oil}) \tag{9}$$

where $\rho_{oil}$ and $\rho_w$ are the density of oil and seawater, respectively.

Thus, compared with the neutrally buoyant state of Petrel-L, the total weight variation of buoyancy unit $B_b$ can be calculated by Equation (10). When $B_b$ is positive, Petrel-L dives in the seawater. On the contrary, it climbs when $B_b$ is negative.

$$B_b = \Delta m_b g = (\Delta m_{bin} + \Delta m_{bout})g = \frac{\Delta m_{bin}}{\rho_{oil}}\rho_w g = (V_b - V_{bN})\rho_w g \tag{10}$$

In the body frame, the position variation of the transferred oil, $\Delta r_{b1}$, and mass variation of seawater inside the rear fairing, $\Delta r_{b2}$, can be expressed as

$$\Delta r_{b1} = \begin{bmatrix} l_{bin} - l_{bout} \\ 0 \\ 0 \end{bmatrix}, \Delta r_{b2} = \begin{bmatrix} l_{bout} \\ 0 \\ 0 \end{bmatrix} \tag{11}$$

where $l_{bin}$ and $l_{bout}$ are the distances of the internal oil tank and the external bladder to the center of buoyancy, respectively.

In this paper, the bladder and the oil tank are considered to be symmetrical relative to the *O-xz*, so the inertia tensor variation, $\Delta J_b$, caused by oil transfer can be expressed as

$$\Delta J_b = (-\Delta m_{bin}(l_{bin} - l_{bout})^2 + \Delta m_b l_{bout}^2)\begin{bmatrix} 0 & 0 & 0 \\ 0 & 1 & 0 \\ 0 & 0 & 1 \end{bmatrix} \tag{12}$$

Petrel-L adjusts its pitch angle and heading by translating and rotating its internal mass block. This paper ignores the effect of the acceleration of the mass block on the glider motion. As shown in Figure 3b, a local coordinate frame, $O_p$-$x_p y_p z_p$, is definite at the center of weight of the mass block, the initial center of which is $r_{p0} = [x_{p0}, 0, z_{p0}]^T$.

In the body frame, when the translation distance and the rotation angle of the internal mass block are $l_p$ and $\varphi_r$, respectively, the movement of the center of weight, $\Delta r_p$, can be expressed as:

$$\Delta r_p = \begin{bmatrix} 1 & 0 & 0 \\ 0 & \cos\varphi_r & -\sin\varphi_r \\ 0 & \sin\varphi_r & \cos\varphi_r \end{bmatrix}\left(r_{p0} + \begin{bmatrix} l_p \\ 0 \\ 0 \end{bmatrix}\right) - r_{p0} = \begin{bmatrix} l_p \\ -z_{p0}\sin\varphi_r \\ z_{p0}(\cos\varphi_r - 1) \end{bmatrix} \tag{13}$$

Influenced by the calibration error and some other factors, there usually exists a rotation angle error, which has an effect on the glide motion. Considering the initial rotation angle error of the mass block, $\Delta\varphi$, $\Delta\mathbf{r}_\mathrm{p}$ can be further expressed by Equation (13).

$$\Delta\mathbf{r}_\mathrm{p} = \begin{bmatrix} l_\mathrm{p} \\ -z_{\mathrm{p}0}\sin(\varphi_\mathrm{r} + \Delta\varphi) \\ z_{\mathrm{p}0}(\cos(\varphi_\mathrm{r} + \Delta\varphi) - 1) \end{bmatrix} \tag{14}$$

The increment of inertia tensor in the body frame is expressed as

$$\Delta J_\mathrm{p} = \begin{bmatrix} \Delta J_{\mathrm{p},xx} & \Delta J_{\mathrm{p},xy} & \Delta J_{\mathrm{p},xz} \\ \Delta J_{\mathrm{p},xy} & \Delta J_{\mathrm{p},yy} & \Delta J_{\mathrm{p},yz} \\ \Delta J_{\mathrm{p},xz} & \Delta J_{\mathrm{p},yz} & \Delta J_{\mathrm{p},zz} \end{bmatrix} \tag{15}$$

where

$$\begin{aligned} \Delta J_{\mathrm{p},xx} &= 0 \\ \Delta J_{\mathrm{p},xy} &= (J_{\mathrm{P}_0,xz} + m_\mathrm{p}(x_{\mathrm{p}0} + l_\mathrm{p})z_{\mathrm{p}0})\sin\varphi_\mathrm{r} \\ \Delta J_{\mathrm{p},xz} &= -(J_{\mathrm{P}_0,xz} + m_\mathrm{p}(x_{\mathrm{p}0} + l_\mathrm{p})z_{\mathrm{p}0})(\cos\varphi_\mathrm{r} - 1) - m_\mathrm{p}l_\mathrm{p}z_{\mathrm{p}0} \\ \Delta J_{\mathrm{p},yy} &= (J_{\mathrm{P}_0,zz} - J_{\mathrm{P}_0,yy})\sin^2\varphi_\mathrm{r} + m_\mathrm{p}l_\mathrm{p}(2x_{\mathrm{p}0} + l_\mathrm{p}) \\ \Delta J_{\mathrm{p},yz} &= 0.5\sin(2\varphi_\mathrm{r})(J_{\mathrm{P}_0,yy} - J_{\mathrm{P}_0,zz} + m_\mathrm{p}z_{\mathrm{p}0}^2) \\ \Delta J_{\mathrm{p},zz} &= (J_{\mathrm{P}_0,yy} - J_{\mathrm{P}_0,zz})\sin^2\varphi_\mathrm{r} + m_\mathrm{p}l_\mathrm{p}(2x_{\mathrm{p}0} + l_\mathrm{p}) \end{aligned} \tag{16}$$

### 2.2.2. Weight, Buoyancy and Righting Moment

The gravity force and the buoyancy of the Petrel-L are $G_0$ and $B_0$, respectively. When the glider is neutrally buoyant, the center of weight, $\mathbf{r}_{G0}$, is

$$\mathbf{r}_{G0} = [0\,0\,z_{G0}]^\mathrm{T} \tag{17}$$

where $z_{G0}$ is the height of the metacenter.

The inertia tensor, $J_0$, of neutrally buoyant Petrel-L in the body frame is

$$J_0 = \begin{bmatrix} J_{0,xx} & 0 & -J_{0,xz} \\ 0 & J_{0,yy} & 0 \\ -J_{0,xz} & 0 & J_{0,zz} \end{bmatrix} \tag{18}$$

When the oil volume inside the oil tank is $V_\mathrm{b}$, the total mass $m$ of Petrel-L is

$$m = m_0 + \Delta m_\mathrm{b} = m_0 + (V_\mathrm{b} - V_{\mathrm{bN}})\rho_\mathrm{w} \tag{19}$$

where $m_0$ is the mass of the UG when it is in the initial neutrally buoyant state.

The center of mass is influenced by the relative motion of the oil and the mass block, the vector of which, $\mathbf{r}_G$, can be expressed as

$$\mathbf{r}_G = \begin{bmatrix} x_G \\ y_G \\ z_G \end{bmatrix} = \frac{m_0\mathbf{r}_{G0} + \Delta m_\mathrm{bin}\Delta\mathbf{r}_{\mathrm{b}1} + \Delta m_\mathrm{b}\Delta\mathbf{r}_{\mathrm{b}2} + m_\mathrm{p}\Delta\mathbf{r}_\mathrm{p}}{m} \tag{20}$$

where $\Delta\mathbf{r}_{\mathrm{b}1}$ and $\Delta\mathbf{r}_{\mathrm{b}2}$ are the displacement variations of center of weight.

Thus, the total inertia tensor, $J$, of Petrel-L can be expressed as

$$J = \begin{bmatrix} J_x & -J_{xy} & -J_{xz} \\ -J_{xy} & J_y & -J_{yz} \\ -J_{xz} & -J_{yz} & J_z \end{bmatrix} = J_0 + \Delta J_\mathrm{b} + \Delta J_\mathrm{p} \tag{21}$$

where $J_x$, $J_y$, and $J_z$ are the rotational inertias relative to the *x*-axis, *y*-axis, and *z*-axis, respectively, $J_{xy}$, $J_{yz}$, and $J_{xz}$ are the products of inertia relative to the plane, *O-xy*, the plane, *O-xy*, and the plane, *O-xy*, respectively.

The weight of Petrel-L $F_G$ and righting moment, $T_G$, can be expressed as

$$F_G = mg \begin{bmatrix} -\sin\theta \\ \cos\theta\sin\varphi \\ \cos\theta\cos\varphi \end{bmatrix} \tag{22}$$

$$T_G = r_G \times F_G = mg \begin{bmatrix} x_G \\ y_G \\ z_G \end{bmatrix} \times \begin{bmatrix} -\sin\theta \\ \cos\theta\sin\varphi \\ \cos\theta\cos\varphi \end{bmatrix} = mg \begin{bmatrix} -z_G\cos\theta\sin\varphi + y_G\cos\theta\cos\varphi \\ -z_G\sin\theta - x_G\cos\theta\cos\varphi \\ y_G\sin\theta + x_G\cos\theta\sin\varphi \end{bmatrix} \tag{23}$$

Thus, the buoyancy force, $F_B$, acting on Petrel-L in the body frame is

$$F_B = B_0 \begin{bmatrix} -\sin\theta \\ \cos\theta\sin\varphi \\ \cos\theta\cos\varphi \end{bmatrix} \tag{24}$$

In the initial neutrally buoyant state, the weight, $G_0$, is equal to the buoyancy $B_0$.

$$G_0 = m_0 g = -B_0 = -\rho_0 V_0 g \tag{25}$$

where $\rho_0$ is the density of seawater at the surface, $V_0$ is the volume of Petrel-L under the barometric pressure.

To sum up, the net buoyancy of Petrel-L can be expressed as

$$F_{BN} = F_G + F_B = \Delta m_b g \begin{bmatrix} -\sin\theta \\ \cos\theta\sin\varphi \\ \cos\theta\cos\varphi \end{bmatrix} \tag{26}$$

### 2.2.3. Viscous Hydrodynamic Force

The viscous hydrodynamic forces, including the forces, $F_{VHV} = [D, SF, L]^T$, and the hydrodynamic moments, $T_{VHV} = [T_{x'}, T_{y'}, T_{z'}]^T$, are caused by the surrounding fluid when Petrel-L is in the state of steady motion.

$$F_{VHV} = \begin{bmatrix} D \\ SF \\ L \end{bmatrix} = \begin{bmatrix} C_D\|V\|^2 \\ C_{SF}\|V\|^2 \\ C_L\|V\|^2 \end{bmatrix} \tag{27}$$

$$T_{VHV} = \begin{bmatrix} T_{x'} \\ T_{y'} \\ T_{z'} \end{bmatrix} = \begin{bmatrix} C_{x'}\|V\|^2 \\ C_{y'}\|V\|^2 \\ C_{z'}\|V\|^2 \end{bmatrix} \tag{28}$$

where drag, *D*, side force, *SF*, and lift, *L*, are the hydrodynamic forces in the velocity frame relative to the *x'*-axis, *y'*-axis, and *z'*-axis, respectively, $T_{x'}$, $T_{y'}$, and $T_{z'}$ are the hydrodynamic moments in the velocity frame relative to the *x'*-axis, *y'*-axis, and *z'*-axis, respectively, $C_D$, $C_{SF}$, and $C_L$ are the effective hydrodynamic force coefficients for drag, side force, and lift, respectively, and $C_{x'}$, $C_{y'}$ and $C_{z'}$ are the effective hydrodynamic moment coefficient, for the moments in the velocity frame relative to the *x'*-axis, *y'*-axis, and *z'*-axis, respectively.

According to the coordinate transformation relation of Petrel-L, the hydrodynamic forces and moments in the body frame can be obtained.

$$F_{VH} = R_B^V F_{VHV} \tag{29}$$

$$T_{VH} = R_B^V T_{VHV} \tag{30}$$

### 2.2.4. Inertial Hydrodynamic Force

The inertial hydrodynamic forces are caused by the surrounding fluid of Petrel-L when the UG accelerates in the seawater, which are determined by the shape of Petrel-L and the density of the fluid. Usually, the inertial hydrodynamic forces are introduced into the dynamic model by the form of the added mass. Considering the symmetry of Petrel-L, the added mass $\lambda_{ij}$ [31] of Petrel-L is

$$\lambda_{ij} = \begin{bmatrix} \lambda_{11} & 0 & 0 & 0 & 0 & 0 \\ 0 & \lambda_{22} & 0 & 0 & 0 & \lambda_{26} \\ 0 & 0 & \lambda_{33} & 0 & \lambda_{35} & 0 \\ 0 & 0 & 0 & \lambda_{44} & 0 & 0 \\ 0 & 0 & \lambda_{35} & 0 & \lambda_{55} & 0 \\ 0 & \lambda_{26} & 0 & 0 & 0 & \lambda_{66} \end{bmatrix} \tag{31}$$

In the body frame, the inertial hydrodynamic forces, $\boldsymbol{F_{IH}}$, and moments, $\boldsymbol{T_{IH}}$, are expressed as

$$\boldsymbol{F_{IH}} = \begin{bmatrix} -\lambda_{11}\dot{u} \\ -\lambda_{22}\dot{v} - \lambda_{26}\dot{r} \\ -\lambda_{33}\dot{w} - \lambda_{35}\dot{q} \end{bmatrix} \tag{32}$$

$$\boldsymbol{T_{IH}} = \begin{bmatrix} -\lambda_{44}\dot{p} \\ -\lambda_{55}\dot{q} - \lambda_{35}\dot{w} \\ -\lambda_{66}\dot{r} - \lambda_{26}\dot{v} \end{bmatrix} \tag{33}$$

### 2.3. Dynamics

In the inertial frame, the momentum, $\boldsymbol{p}$, and the moment of momentum, $\boldsymbol{L}$, relative to O are

$$\boldsymbol{p} = m\boldsymbol{V_G} = m(\boldsymbol{V} + \dot{\boldsymbol{r}}_G) = m(\boldsymbol{V} + \boldsymbol{\Omega} \times \boldsymbol{r}_G) \tag{34}$$

$$\boldsymbol{L} = \boldsymbol{J}\boldsymbol{\Omega} + m\boldsymbol{r}_G \times \boldsymbol{V} \tag{35}$$

where $\boldsymbol{V_G} = \boldsymbol{V} + \boldsymbol{\Omega} \times \boldsymbol{r_G}$ is the velocity of the center of mass.

In the body frame, the total external forces and the external moments acted on the Petrel-L are

$$\boldsymbol{F} = \boldsymbol{F_G} + \boldsymbol{F_B} + \boldsymbol{F_{VH}} + \boldsymbol{F_{IH}} \tag{36}$$

$$\boldsymbol{T} = \boldsymbol{T_G} + \boldsymbol{T_{VH}} + \boldsymbol{T_{IH}} \tag{37}$$

The dynamic equations can be obtained according to the momentum theorem and the moment of momentum theorem [20].

$$\frac{\mathrm{d}\boldsymbol{p}}{\mathrm{d}t} = \frac{\widetilde{\mathrm{d}}\boldsymbol{p}}{\mathrm{d}t} + \boldsymbol{\Omega} \times \boldsymbol{p} = m\left[\frac{\widetilde{\mathrm{d}}\boldsymbol{V}}{\mathrm{d}t} + \frac{\widetilde{\mathrm{d}}\boldsymbol{\Omega}}{\mathrm{d}t} \times \boldsymbol{r}_G + \boldsymbol{\Omega} \times (\boldsymbol{V} + \boldsymbol{\Omega} \times \boldsymbol{r}_G)\right] = \boldsymbol{F} \tag{38}$$

$$\frac{\mathrm{d}\boldsymbol{L}}{\mathrm{d}t} = \frac{\widetilde{\mathrm{d}}\boldsymbol{L}}{\mathrm{d}t} + \boldsymbol{\Omega} \times \boldsymbol{L} + \boldsymbol{V} \times \boldsymbol{p} = \boldsymbol{J}\frac{\widetilde{\mathrm{d}}\boldsymbol{\Omega}}{\mathrm{d}t} + m\boldsymbol{r}_G \times \frac{\widetilde{\mathrm{d}}\boldsymbol{V}}{\mathrm{d}t} + \boldsymbol{\Omega} \times (\boldsymbol{J}\boldsymbol{\Omega} + m\boldsymbol{r}_G \times \boldsymbol{V}) + \boldsymbol{V} \times m(\boldsymbol{\Omega} \times \boldsymbol{r}_G) = \boldsymbol{T} \tag{39}$$

where $\frac{\mathrm{d}}{\mathrm{d}t}$ is the time derivative of the vector in the inertia frame, and $\frac{\widetilde{\mathrm{d}}}{\mathrm{d}t}$ is the time derivative of the vector in the body frame.

The decomposition equations can be expressed by Equation (39) according to the projection along the *O-x* axis, *O-y* axis, and *O-z* axis in the body frame.

$$\begin{cases} m\left[\dot{u} - vr + wq - x_G\left(q^2 + r^2\right) + y_G\left(pq - \dot{r}\right) + z_G\left(pr + \dot{q}\right)\right] = F_x \\ m\left[\dot{v} - wp + ur - y_G\left(r^2 + p^2\right) + z_G\left(rq - \dot{p}\right) + x_G\left(qp + \dot{r}\right)\right] = F_y \\ m\left[\dot{w} - uq + vp - z_G\left(p^2 + q^2\right) + x_G\left(pr - \dot{q}\right) + y_G\left(rq + \dot{p}\right)\right] = F_z \\ J_x\dot{p} - \left(J_y - J_z\right)qr + J_{xy}\left(pr - \dot{q}\right) + J_{yz}\left(r^2 - q^2\right) - J_{xz}\left(\dot{r} + pq\right) + my_G\left(\dot{w} + pv - qu\right) \\ \quad -mz_G\left(\dot{v} + ru - pw\right) = T_x \\ J_y\dot{q} - \left(J_z - J_x\right)rp + J_{yz}\left(qp - \dot{r}\right) + J_{xz}\left(p^2 - r^2\right) - J_{xy}\left(\dot{p} + qr\right) + mz_G\left(\dot{u}z + qw - rv\right) \\ \quad -mx_G\left(\dot{w} + pv - qu\right) = T_y \\ J_z\dot{r} - \left(J_x - J_y\right)pq + J_{xz}\left(rq - \dot{p}\right) + J_{xy}\left(q^2 - p^2\right) - J_{yz}\left(\dot{q} + rp\right) + mx_G\left(\dot{v} + ru - pw\right) \\ \quad -my_G\left(\dot{u} + qw - rv\right) = T_z \end{cases} \quad (40)$$

where $Fx$, $Fy$, $Fz$ and $Tx$, $Ty$, $Tz$ are projections of the external force principal vector, $\boldsymbol{F}$, and the principal moment, $\boldsymbol{T}$, along the three axes.

Equations (2), (4) and (40) make up the dynamic model of the glider, which is used for simulating the glider motion. In this study, we use the MATLAB software to solve the dynamics, and the numerical method used is ode45, which is widely used for solving the ordinary differential equation. In the simulation, the initial values of $\boldsymbol{V}$, $\boldsymbol{\Omega}$, $\boldsymbol{b}$, and $\boldsymbol{\eta}$ are set as 0, and the variations of them can be obtained by importing the input parameters, including the oil volume $V_{bN}$, inside the internal oil tank and the position of the mass pack, consisting of the translation distance, $l_p$, and the rotation angle, $\varphi_r$.

## 3. Trajectory Error Analysis and Identification Strategy

### 3.1. Trajectory Error Analysis

In this paper, we ignore the effect of the ocean current on the trajectory. Usually, Petrel-L glides in the vertical plane when the initial rotation angle error, $\Delta\varphi$, of mass block in Equation (13) is $0°$. However, caused by the trim error and calibration error, the initial rotation angle error, $\Delta\varphi$, of mass block exists, which leads to the deviation of heading of Petrel-L. To reduce the energy consumed by frequent heading adjustments, the heading, $\Delta\psi$, is set as $15°$ in the practical engineering, and Petrel-L begins to adjust its heading when the deviation of its heading to the planned heading, $\psi_T$, is larger than $\Delta\psi$. In the process of deviation of heading, Petrel-L moves in a spiral motion in the three-dimensional space, which causes Petrel-L to deviate from the vertical plane and results in the trajectory error, $\Delta Y$, shown in Figure 4. Due to the lack of an underwater positioning sensor carried by Petrel-L, it can only adjust the heading instead of its underwater position.

Compared with the process of the deviation of heading, the adjustment of the heading is relatively quick because of a larger rotation angle of the internal mass block, $\varphi_r$, a process which is ignored in this study. Although the heading has been adjusted as the planned heading, Petrel-L cannot return to the planned trajectory, and the trajectory error is generated. Moreover, as the number of the heading adjustments increase, the trajectory error gradually increases, and Petrel-L keeps away from the planned trajectory, which is disadvantageous for its application in ocean observation.

With the dynamic model established in Section 2, the effect of the input parameters on the trajectory error can be analyzed. Figures 5 and 6 show the trajectory errors of Petrel-L in dive motion under the different pitch angles, net buoyancies, and rotation angle errors, respectively, when the deviation of its heading gradually changes from $0°$ to $15°$, which indicates that the trajectory error increases with the pitch angle and net buoyancy and decreases with the rotation angle error. Thus, the trajectory error is more dependent on the velocity of the UG because the UG moves a longer distance under the same deviation of its heading. The heading of the UG with a larger rotation angle error will change faster, and the time consumed for the deviation of heading from $0°$ to $15°$ is shorter, so the trajectory error is smaller.

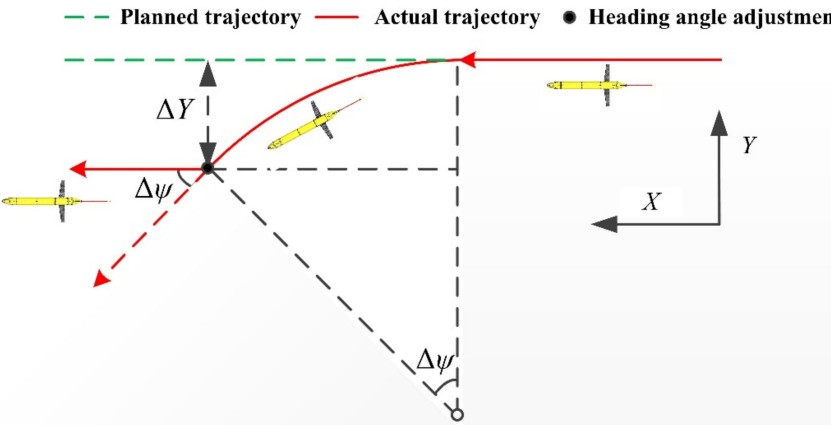

**Figure 4.** Trajectory error caused by heading adjustment.

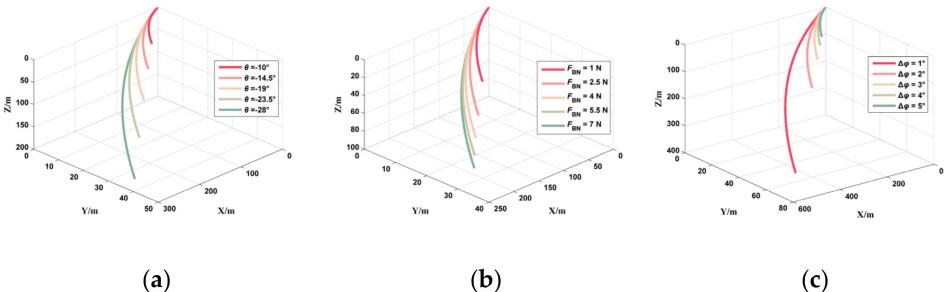

(a)                                 (b)                                 (c)

**Figure 5.** Three-dimensional trajectory of Petrel-L under different pitch angles (**a**), net buoyancies (**b**), and rotation angle errors (**c**).

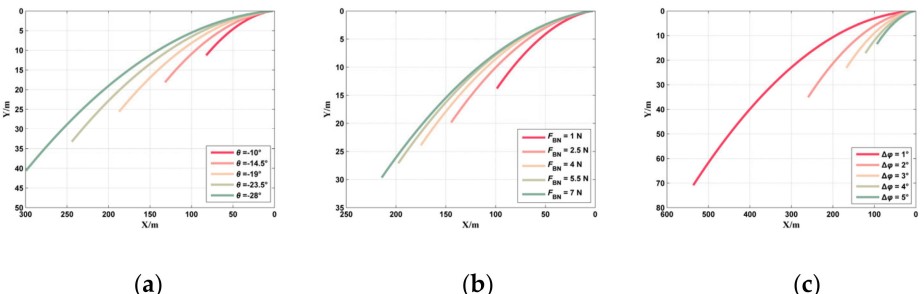

(a)                                 (b)                                 (c)

**Figure 6.** Two-dimensional trajectory of Petrel-L under different pitch angles (**a**), net buoyancies (**b**), and rotation angle errors (**c**).

To obtain the trajectory error of Petrel-L for trajectory optimization, an identification strategy needs to be proposed based on the dynamic model and glider datasets in the sea trial. Due to the strong nonlinearity of the dynamic model, the relationship between the input parameters (net buoyancy, pitch angle, and rotation angle errors) and the output parameters (vertical velocity, $Z'$, horizontal velocity, $V_{XY}$, and heading angular velocity $\psi'$) cannot be explicitly expressed by a function, which makes it difficult to directly identify the trajectory error by the data of Petrel-L. Thus, the approximate models are established to express these relationships replacing the dynamic model. The establishment processes of the approximate models are shown as follows.

(1)  Design space: the design space of the approximate models needs to be determined according to the engineering practice, which is shown in Table 1.

**Table 1.** Design space of approximate models.

| Parameter | Value Range | Parameter Type |
|-----------|-------------|----------------|
| $|F_{BN}|$ | 1–9 N | Real |
| $\Delta\varphi$ | 0–10° | Real |
| $|\theta|$ | 10~30° | Real |

(2)  Data collection: The optimal Latin hypercube method [32] is adopted the design space which can ensure a uniform distribution of the sampling points. The initial sample size is 400. Considering the symmetry of the glide motion, we use the approximated dynamic model and record from the pitch, heading, and depth sensors to estimate the trajectory error while the vehicle dives.

(3)  Establishment and verification: Seventy-five percent of the datasets are used to establish the approximate models in which the response surface method [33] is used. The surplus 25 percent of the datasets are used to verify the accuracies of the approximate models. If the correlation coefficients are smaller than 0.99, we will increase the sample number and repeat the processes (1)–(3) until the accuracies meet the requirements.

The approximate models established by 300 datasets are shown in Figures 7–9, which indicate the effect of the design variables on the output parameters. Figures 7 and 8 indicate that the vertical velocity, $Z'$, and the horizontal velocity, $V_{XY}$, increase with the net buoyancy, $F_{BN}$, and the pitch angle, $\theta$, and vary little with the rotation angle error, $\Delta\varphi$. Figure 9 indicates that the heading angular velocity, $\psi'$, increases with the net buoyancy, $F_{BN}$, the pitch angle, $\theta$, and the rotation angle error, $\Delta\varphi$.

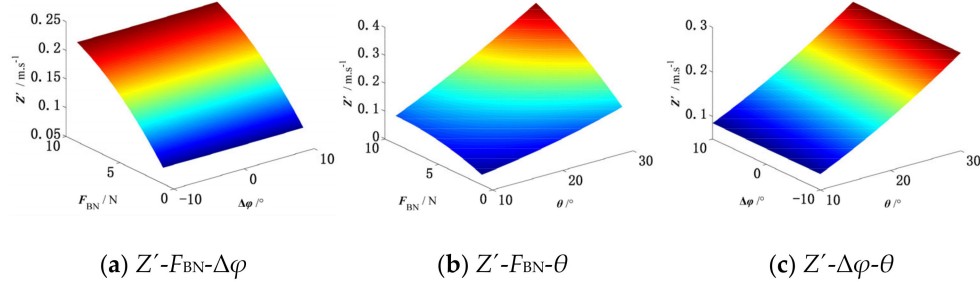

**(a)** $Z'$-$F_{BN}$-$\Delta\varphi$      **(b)** $Z'$-$F_{BN}$-$\theta$      **(c)** $Z'$-$\Delta\varphi$-$\theta$

**Figure 7.** Approximate model of vertical velocity.

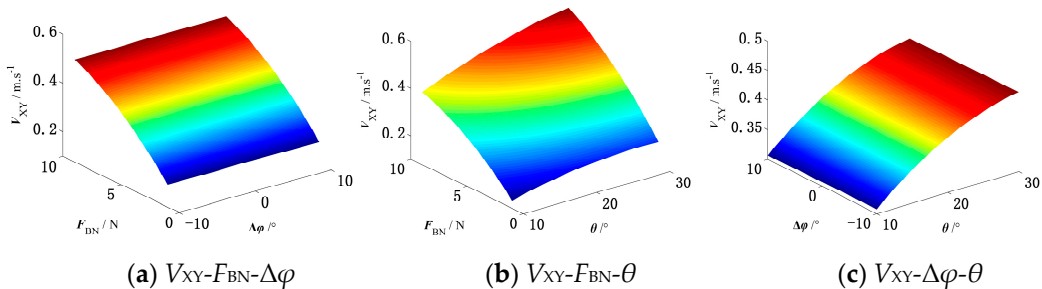

**(a)** $V_{XY}$-$F_{BN}$-$\Delta\varphi$      **(b)** $V_{XY}$-$F_{BN}$-$\theta$      **(c)** $V_{XY}$-$\Delta\varphi$-$\theta$

**Figure 8.** Approximate model of horizontal velocity.

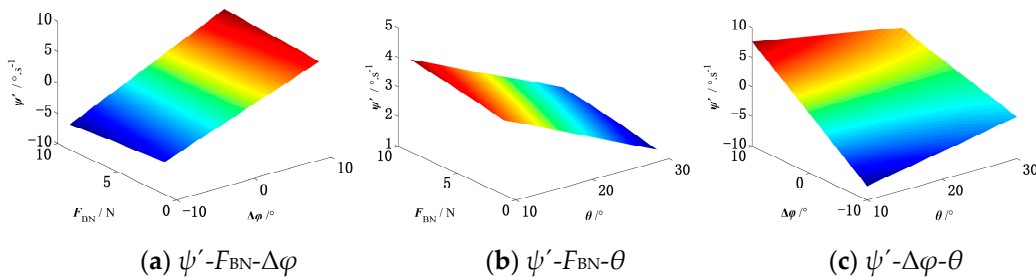

**(a)** $\psi'$-$F_{\text{BN}}$-$\Delta\varphi$  **(b)** $\psi'$-$F_{\text{BN}}$-$\theta$  **(c)** $\psi'$-$\Delta\varphi$-$\theta$

**Figure 9.** Approximate model of heading angular velocity.

Figure 10a–c show the accuracy of the approximate models. By verification with the 100 datasets, the correlation coefficients of these three approximate models are 0.995, 0.996, and 0.999 respectively, which indicate that the approximate models are effective to be used in the identification of the trajectory error. Equations (40)–(42) are the specific expressions of approximate models in the dive motion, and the approximate models in the climb motion can be obtained from them.

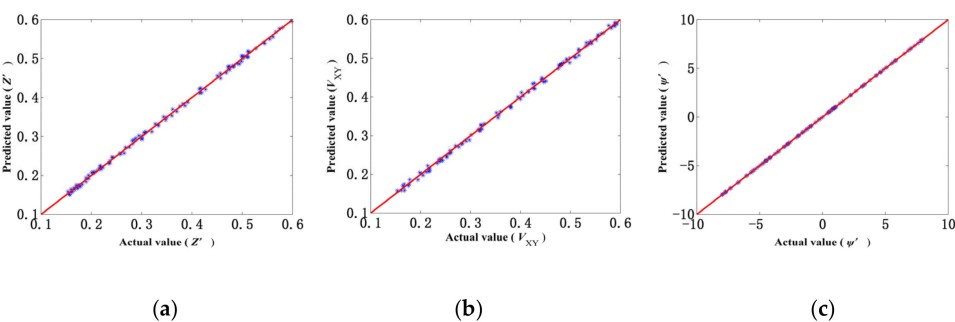

**(a)**  **(b)**  **(c)**

**Figure 10.** Accuracy of approximate models ($Z'$**(a)**, $V_{\text{XY}}$**(b)** and $\psi'$**(c)** ).

$$V_{\text{XY}} = -0.022 + 0.012|\theta| - 1.07 \times 10^{-4}\Delta\varphi + 0.052|F_{\text{BN}}| - 2.27 \times 10^{-4}|\theta|^2 + 7.17 \times 10^{-7}\Delta\varphi^2$$
$$-2.4 \times 10^{-3}F_{\text{BN}}^2 + 7.42 \times 10^{-7}|\theta|\Delta\varphi + 6.8 \times 10^{-4}|\theta F_{\text{BN}}| + 1.36 \times 10^{-5}\Delta\varphi|F_{\text{BN}}| \tag{41}$$

$$Z' = 7.79 \times 10^{-3} + 4.1 \times 10^{-5}|\theta| - 3.91 \times 10^{-5}\Delta\varphi + 8.38 \times 10^{-3}|F_{\text{BN}}| + 1.02 \times 10^{-4}|\theta|^2 + 4.58 \times 10^{-6}\Delta\varphi^2$$
$$-1.05 \times 10^{-3}F_{\text{BN}}^2 + 7.94 \times 10^{-7}|\theta|\Delta\varphi + 1.02 \times 10^{-3}|\theta F_{\text{BN}}| + 2.59 \times 10^{-6}\Delta\varphi|F_{\text{BN}}| \tag{42}$$

$$\psi' = -7.54 \times 10^{-3} - 8.14 \times 10^{-4}|\theta| + 0.92\Delta\varphi + 0.014|F_{\text{BN}}| - 2.65 \times 10^{-5}|\theta|^2$$
$$+6.76 \times 10^{-5}\Delta\varphi^2 - 1.58 \times 10^{-3}F_{\text{BN}}^2 - 0.023|\theta|\Delta\varphi + 1.39 \times 10^{-4}|\theta F_{\text{BN}}| + 0.015\Delta\varphi|F_{\text{BN}}| \tag{43}$$

*3.2. Identification Strategy of Trajectory Error*

In this section, we try to identify the trajectory error of Petrel-L by the approximate models established above and its time series datasets in the sea trial. In Equations (41)–(43), some variables can be directly measured by the sensors carried by Petrel-L, including the pitch angle, $\theta$, and the heading angular velocity, $\psi'$, measured by a TCM3 sensor and vertical velocity, $Z'$, measured by a pressure sensor. Other variables are unknown and need to be identified.

As shown in Figure 4, the trajectory error, $\Delta Y$, can be deduced by the deviation of the heading, $\Delta\psi$, and the arc length is calculated by the vertical velocity and heading angular velocity, $\psi'$, shown as

$$\Delta Y = \int_{t_1}^{t_2} V_{\text{XY}} t \sin\psi_{\text{T}} \mathrm{d}t \tag{44}$$

where $t_1$ and $t_2$ are the start time and end time of the datasets, respectively, $t$ is the time of datasets, and $\psi_{\text{T}}$ is the target heading of Petrel-L.

According to Equations (41)–(44), the following identification strategies of trajectory error are generated.

(1) By data processing of the Petrel-L when its heading gradually changes in the sea trial, the time series datasets, including the pitch angle, $\theta$, vertical velocity, $Z'$, and heading angular velocity, $\psi'$, can be obtained.

(2) By importing the dataset $[\theta \ Z' \ \psi']$ obtained in (1) into Equations (42) and (43), the net buoyancy, $F_{BN}$, and rotation angle error, $\Delta\varphi$, can be obtained by solving the equation set.

(3) By importing the dataset $[\theta \ F_{BN} \ \Delta\varphi]$ into Equation (41), the vertical velocity, $V_{XY}$, can be calculated.

(4) By importing the vertical velocity, $V_{XY}$, into Equation (44), the trajectory error, $\Delta Y$, under the deviation of the heading, $\Delta\psi$, can be obtained, and the total trajectory error can be obtained by repeating processes (1)–(4).

## 4. Trajectory Optimization Strategy and Discussion

A trajectory error, $\Delta Y$, is caused when the rotation-steering angle has an error, $\Delta\varphi$, after Petrel-L has adjusted its heading. The trajectory error, $\Delta Y$, generates when the rotation angle error, $\Delta\varphi$, of the mass block exists, and Petrel-L adjusts the heading. According to the above analysis, the trajectory error, $\Delta Y$, can be identified by the approximate models and the datasets of Petrel-L. To make Petrel-L return to the planned trajectory and travel in the vertical plane, a trajectory optimization strategy can be proposed with a given trajectory error, $\Delta Y$, the processes of which are shown as follows.

(1) Petrel-L begins to adjust its heading when the deviation of the heading, $\Delta\psi$, is larger than 15°. To compensate the trajectory error, $\Delta Y$, a heading adjustment, $\psi_o$, is required, shown by the red dotted lines in Figure 11. Thus, the distance travelled by Petrel-L is

$$L_d = \Delta Y / \sin \psi_o \tag{45}$$

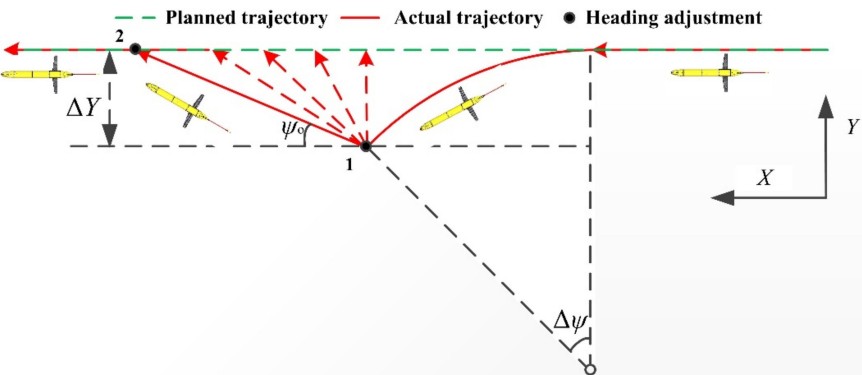

**Figure 11.** Trajectory optimization strategy.

(2) With the identification strategy proposed in Section 3, the rotation angle error, $\Delta\varphi$, of the mass block can be obtained. After the heading adjustment 1, a rotation angle, $-\Delta\varphi$, of the mass block needs to compensate for Petrel-L to realize glide motion in the vertical plane instead of the spiral motion in the three-dimensional space.

(3) With the compensation of the rotation angle, $-\Delta\varphi$, of the mass block, the $\Delta\varphi$ in Equations (41) and (42) can be set as 0, and the heading angular velocity, $\psi'$, in Equation (43) is close to 0. Thus, Equations (46) and (47) can be obtained.

(4) With Equation (47) and the pitch angle of Petrel-L, the real-time net buoyancy can be calculated. by importing which into Equation (46) the horizontal velocity $V_{XY}$ can be calculated.

(5) The distance travelled by Petrel-L can be obtained by the time integration of horizontal velocity, $V_{XY}$, and the heading need, be adjusted to the planned heading when the

distance travelled by Petrel-L reaches $L_\mathrm{d}$. Finally, Petrel-L will continue to glide in the planned trajectory.

$$V_{\mathrm{XY}} = -0.022 + 0.012|\theta| + 0.052|F_{\mathrm{BN}}| - 2.27 \times 10^{-4}|\theta|^2 - 2.4 \times 10^{-3}F_{\mathrm{BN}}{}^2 + 6.8 \times 10^{-4}|\theta F_{\mathrm{BN}}| \tag{46}$$

$$\begin{aligned} Z' &= 7.79 \times 10^{-3} + 4.1 \times 10^{-5}|\theta| + 8.38 \times 10^{-3}|F_{\mathrm{BN}}| + 1.02 \times 10^{-4}|\theta|^2 \\ &\quad - 1.05 \times 10^{-3}F_{\mathrm{BN}}{}^2 + 1.02 \times 10^{-3}|\theta F_{\mathrm{BN}}| \end{aligned} \tag{47}$$

Although the trajectory of Petrel-L can be optimized and adjusted using the above strategy to improve the trajectory accuracy, a gliding range loss of Petrel-L caused by the trajectory error and trajectory optimization strategy will be generated. In addition, the observation distance of Petrel-L along the planned trajectory becomes shorter. Thus, considering the gliding range loss and the observation distance loss, the selection of the heading, $\psi_o$, needs to be discussed.

As shown in Figure 11, the gliding range loss can be obtained by the difference value of the actual trajectory and its projection on the planned trajectory, shown as

$$\begin{aligned} R_{\mathrm{loss}} &= \Delta Y / (1 - \cos \Delta\psi)\Delta\psi(\mathrm{rad}) + \Delta Y / \sin \psi_o \\ &\quad - \Delta Y / (1 - \cos \Delta\psi)\sin \Delta\psi - \Delta Y / \sin \psi_o \cos \psi_o \end{aligned} \tag{48}$$

The observation distance loss along the planned trajectory is calculated as

$$L_{\mathrm{loss}} = \Delta Y / (1 - \cos \Delta\psi)\sin \Delta\psi + \Delta Y / \sin \psi_o \cos \psi_o \tag{49}$$

The selection range of the heading, $\psi_o$, is 0–90°, and the deviation of the heading, $\Delta\psi$, is 15° for Petrel-L. Figure 12a,b show the gliding range loss and the observation distance loss with the heading, $\psi_o$, respectively. As shown in Figure 12a, the gliding range loss increases with the heading, $\psi_o$, the trend of which is basically linear. However, the observation distance loss rapidly decreases with the heading, $\psi_o$, when the heading, $\psi_o$, is smaller than around 10° and varies little when it is larger than 10°. By contrasting Figure 12a,b, the conclusion can be drawn that the gliding range loss is far less than the observation distance loss.

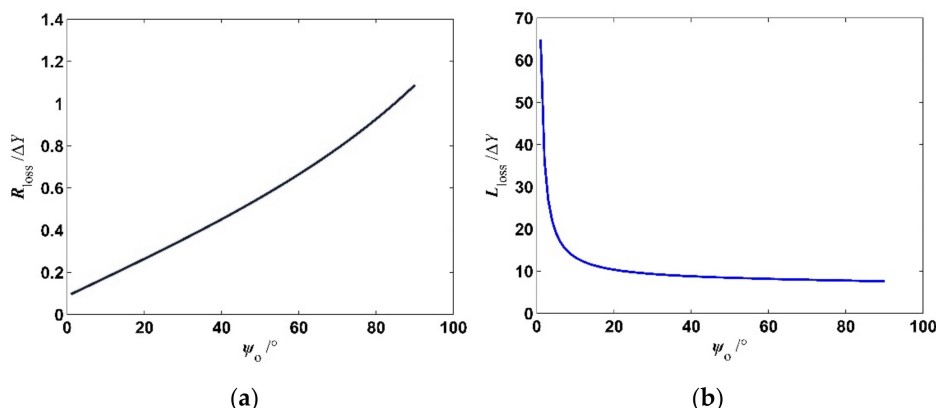

(a)  (b)

**Figure 12.** Gliding range loss (**a**) and observation distance loss (**b**).

In practical engineering, the heading, $\psi_o$, can be determined by the specific observation missions. In some long-term observation missions, a smaller heading, $\psi_o$, is suggested to realize a smaller gliding range loss. However, some observation missions, such as the networking observation mission with multiple gliders, require a high trajectory accuracy, and a larger heading, $\psi_o$, is suggested for obtaining a smaller observation distance loss. Generally, a heading, $\psi_o$, around 10° has a comprehensive performance for realizing a smaller gliding range loss and observation distance loss.

## 5. Test Verification

To verify the effectiveness of the identification strategy proposed in this paper, we adopt some datasets of Petrel-L from 2018 in the South China Sea, the position of which is shown in Figure 13. Considering the effect of the ocean current on the trajectory error, the datasets of some profiles with lower ocean currents are selected. We choose two continuous 1000 m depth profiles of Petrel-L with the same parameter settings except for the rotation angle error, $\Delta\varphi$, of the mass block. The rotation angle error, $\Delta\varphi$, of the mass block in these two profiles are set as $0°$ and $−7°$, respectively. By using the datasets of Petrel-L, the approach proposed in this paper can be carried out to obtain the theoretical position and trajectory error of the Petrel-L at the sea surface after the profile. By contrasting the trajectory error in the sea trial and the identification results, the effectiveness of the identification strategy proposed in this paper can be verified. Moreover, after compensating the rotation angle, $−\Delta\varphi$, of the mass block, the trend of heading can also be contrasted with that before compensation, which can verify the correctness of the approximate models. The identification of the trajectory error is carried out with the software, MATLAB, in a computer with the Windows 10 system and a RAM of 8G, and the process of identification consumes about 1 min.

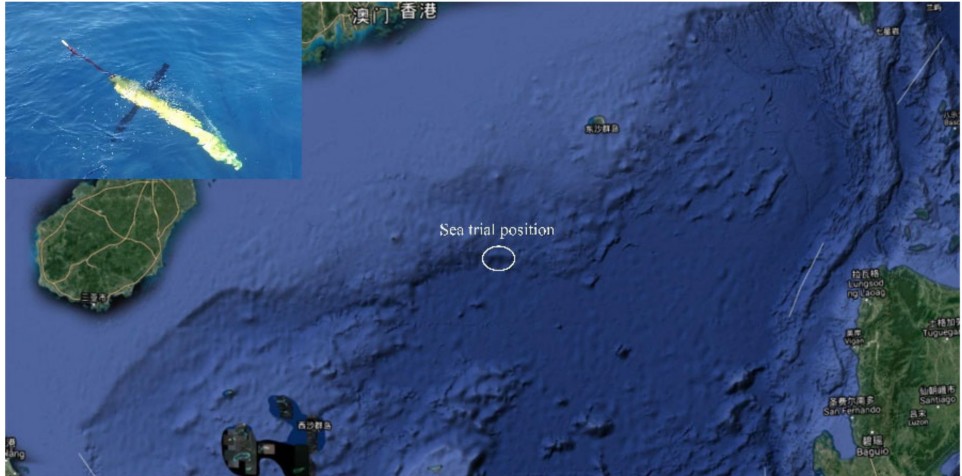

**Figure 13.** Sea trial position.

Figures 14 and 15 show the datasets of Petrel-L in the dive motion and the climb motion, which will be used in the identification strategy of trajectory error. As shown in Figures 14c and 15c, the heading in the dive motion is steady, which indicates that the rotation angle error, $\Delta\varphi$, of the mass block is small enough in the dive motion. However, there exists a larger rotation angle error, $\Delta\varphi$, of the mass block in the climb motion, which causes Petrel-L to frequently adjust its heading. This large difference is caused by the asymmetry of Petrel-L along the *O-xz* plane, which may be caused by the problems of assembly error, biofouling, or corrosion. In the verification, the actual position and the theoretical position of Petrel-L at the sea surface can be purely contrasted due to the lack of the real-time location of Petrel-L. Due to the accumulation of the trajectory errors in dive motion and climb motion, the datasets with the above large difference in dive motion and climb motion can better verify the approach proposed in this paper.

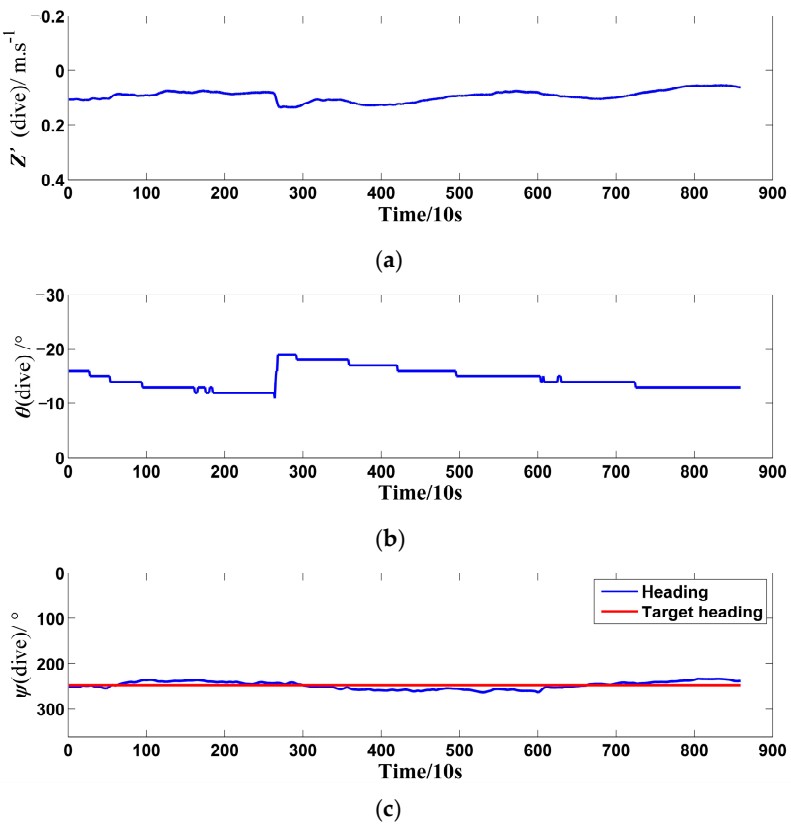

**Figure 14.** Datasets of Petrel-L in dive motion: (**a**) vertical velocity, (**b**) pitch angle, (**c**) heading.

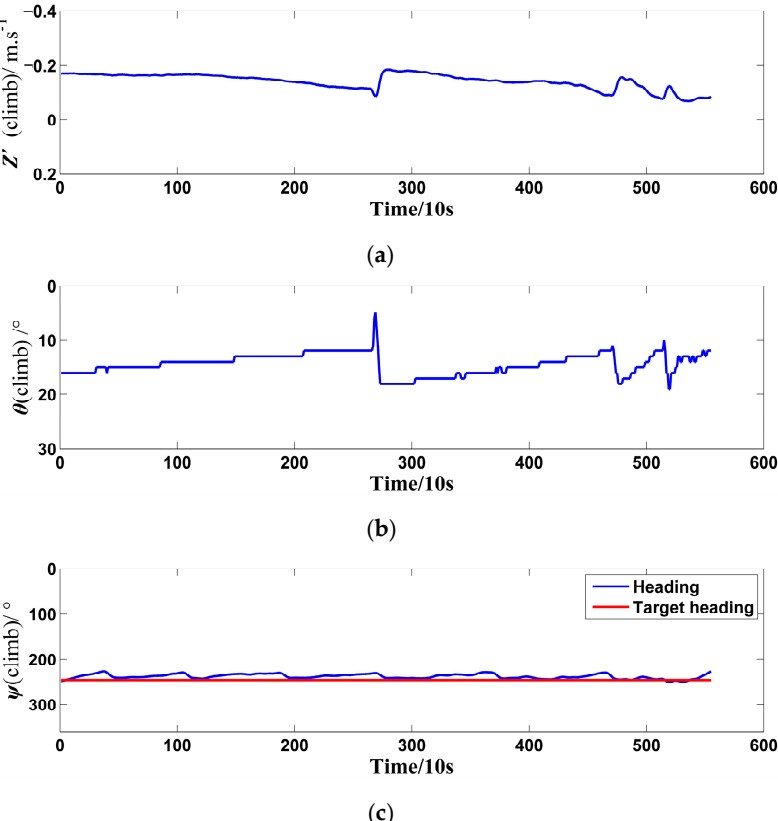

**Figure 15.** Datasets of Petrel-L in climb motion: (**a**) vertical velocity, (**b**) pitch angle, (**c**) heading.

In the dive motion, Petrel-L has adjusted its heading for only one time from 6010 s to 6100 s, the process of which is ignored, and other datasets can be used in the identification. According to the identification strategy in Section 3.2, the average rotation angle error, $\Delta\varphi$, is 0.34°, and the identification results are shown in Figure 16. As shown in Figure 16b, the trajectory error of Petrel-L at the end of the dive motion is −17.17 m.

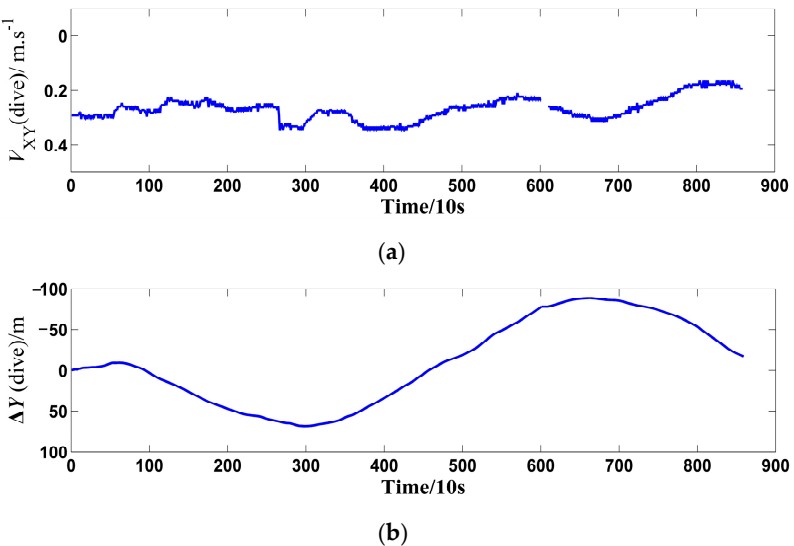

(a)

(b)

**Figure 16.** Identification results of Petrel-L in dive motion: (**a**) horizontal velocity and (**b**) trajectory error.

In the climb motion, Petrel-L has adjusted its heading for only six times, the process of which is ignored, and other datasets can be used in the identification. According to the identification strategy in Section 3.2, the average rotation angle error, $\Delta\varphi$, is 6.72°, and the identification results are shown in Figure 17. As shown in Figure 17b, the trajectory error of Petrel-L in the dive motion is −274.74 m.

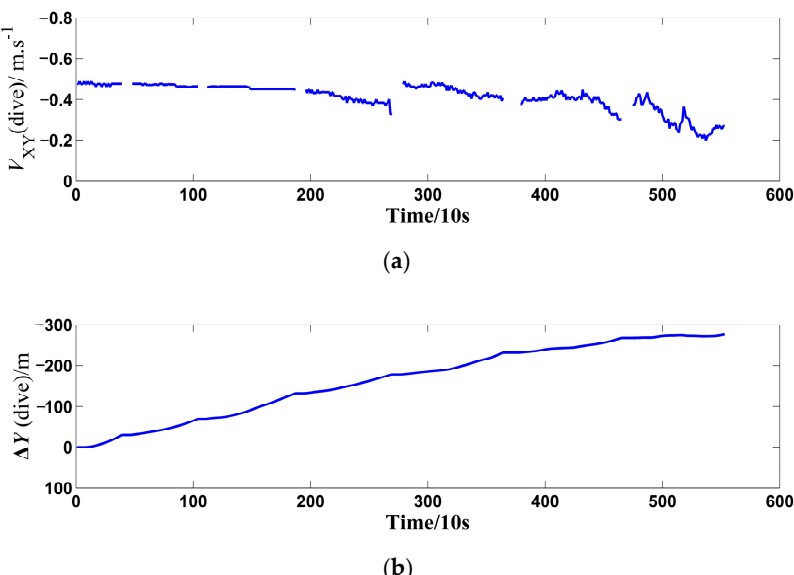

(a)

(b)

**Figure 17.** Identification results of Petrel-L in dive motion: (**a**) horizontal velocity and (**b**) trajectory error.

To sum up, the total estimated trajectory error, $\Delta Y$, from the dive motion and the climb motion is −292 m, which is reasonably close to the measured trajectory error of around

−379 m in the sea trial. Except the influence of the ocean current, the unsteady motion processes of Petrel-L, such as the heading adjustment and the buoyancy adjustment, may also influence the identification accuracy.

As shown in Figure 18, after giving a compensation of −7° of rotation angle error, the heading of Petrel-L varies little in the climb motion and does not need to be frequently adjusted anymore, which verifies the correctness of the identification results.

To sum up, by contrasting the theoretical trajectory error obtained by the approach proposed in this paper and the actual trajectory error, the identification strategy and the identification results are preliminarily verified. Thus, the underwater trajectory of Petrel-L can be adjusted by our approach in the lack of the underwater positioning sensor. To implement the approach in practical engineering, the relevant procedure can be written in the control system of Petrel-L to explore the efficiency of the approach. However, the approach proposed in this paper does not consider the effect of the ocean current on the motion of the UG, which has a certain influence on the trajectory of the UG.

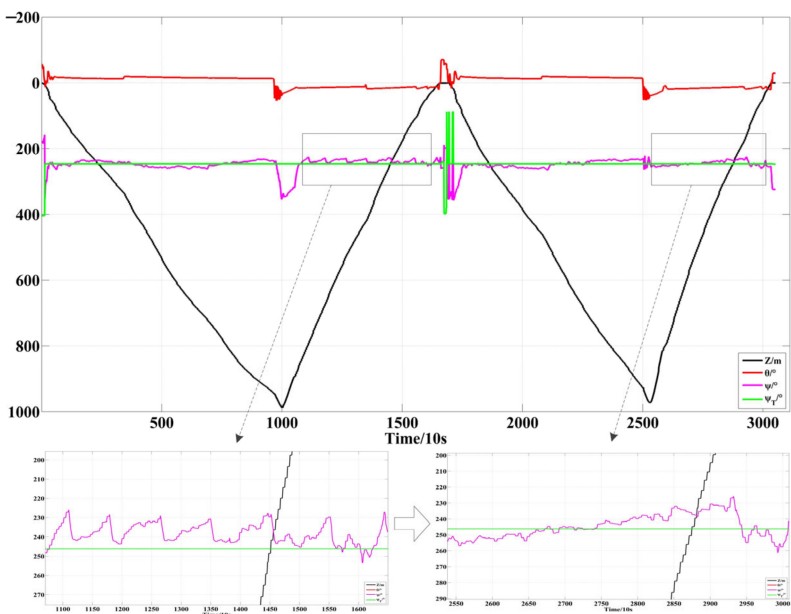

**Figure 18.** Parameter variation caused by compensation of rotation angle error.

## 6. Conclusions

In this paper, a dynamic model of Petrel-L is first established to carry out the trajectory error analysis. Then, the approximate models are established based on the dynamic model, with which an identification strategy of trajectory error for Petrel-L is proposed to obtain the real-time error. To reduce the trajectory error, we propose a trajectory optimization strategy while considering the gliding range loss and observation distance loss. Finally, the error of 87 m between the identification result and the actual trajectory error preliminarily verifies the approach proposed in this paper.

The uniqueness and contribution of this study are fourfold. First, the effect of the heading adjustment on the trajectory error of the Petrel-L glider is studied with the dynamic model. Second, the identification strategy is presented for the first time to obtain the deviation distance of UGs. Third, to improve the trajectory accuracy of UGs, the trajectory optimization schemes are contrasted while considering gliding range loss and observation distance loss. Forth, the proposed method is verified by the datasets of Petrel-L in the sea trial.

For future works to adequately verify the approach proposed in this paper, an underwater positioning device will be integrated by Petrel-L to obtain the real-time position of Petrel-L, and the comparison can be carried out between the theoretical real-time position and actual real-time position. In addition, the ocean current will be considered in the

analysis of the trajectory error of underwater gliders, which has a certain influence on its trajectory.

**Author Contributions:** M.Y.: methodology, writing—original draft preparation. Y.W.: Conceptualization, supervision, project administration. Y.L.: data curation, visualization. Y.S.: software. S.Y.: funding acquisition, writing—review and editing. All authors have read and agreed to the published version of the manuscript.

**Funding:** This work was jointly supported by the National Natural Science Foundation of China (Grant Nos. 51721003 and 11902219); Natural Science Foundation of Tianjin City (Grant No. 18JCJQJC46400); and Aoshan Talent Cultivation Program (Grant No. 2017ASTCP-OE01) of Pilot National Laboratory for Marine Science and Technology (Qingdao).

**Institutional Review Board Statement:** Not applicable.

**Informed Consent Statement:** Not applicable.

**Conflicts of Interest:** The authors declare no conflict of interest.

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
