# Peer review of "A Novel Method of Trajectory Optimization for Underwater Gliders Based on Dynamic Identification"

_jmse, doi:10.3390/jmse10030307_

Round 1

Reviewer 1 Report

This paper discusses a novel and potentially effective method of improving the steering of the Petrel-L underwater glider and other gliders that use an internally-rotated mass to control the heading angle. The approach is a form of model-based predictive control that approximates the full non-linear equations of motion with quadratic equations derived from a statistical technique, Response Surface Methodology, using the Latin Squares experimental design technique. This novel approach relies on accurate dynamic model parameters from multiple previous studies of the glider. The authors demonstrate its effectiveness in simulation using historical trials data. If it performs well in sea trials, it will be a valuable approach, because it does not rely on external sensors to calculate the necessary adjustments to actuator settings. However, the approach must be verified in sea trials and it is not clear from this paper whether this has been done, or whether there are plans to do so. The particular example that is chosen to demonstrate the approach has a small steering rotation angle error when the vehicle is diving, but a much larger error when it is climbing. The paper does not discuss how this large difference originated. It is possible that there are problems in the glider that would make the adjustment of steering rotation angle less effective than the simulation suggests. This should be discussed.

The section on prior work is limited to Chinese authors. There have been many studies by other authors around the world, but only one is mentioned in this article. If prior papers have done a more comprehensive survey, this should be indicated. Otherwise, it would be appropriate to discuss other approaches by Leonard, Mahmoudian, Woolsey and their co-workers. In particular, Mahmoudian is working with gliders that have a similar control system. On the other hand, the abstract does not indicate what is novel about the approach adopted in this paper. It says that an identification strategy is discussed, but not what is different about it from other approaches, which are largely based on linearisation, rather than a form of statistical approximation.

The content is generally of high quality, but there are many minor issues with the English language presentation and there are some paragraphs that are very difficult to understand in their current form. An edited document is attached that indicates how the authors could potentially change their expression to improve it. A few equations may contain errors and there are several places where symbols have not been defined, or where unusual symbols are being used. These should be fixed and all equations should be checked to ensure that they are correct. 

Some of the figures could be improved by changing the scale to make small quantities more obvious.

In summary, this paper rewards the effort of the reader, but it could be improved (as discussed in the attached document) with more explanation to make the content clearer. Further studies involving the effects of currents would be of interest to the scientific community. Sea trials showing how well the proposed method works would greatly improve this paper, but if they are not available another paper is justified.

Reviewer 2 Report

This work presents a dynamic model of Petrel-L to carry out the trajectory error analysis. The approximate models are established based on the dynamic model, and an identification strategy of trajectory error for Petrel-L is proposed to obtain the real-time error. A trajectory optimization strategy is proposed for considering the gliding range loss and observation distance loss. The identification strategy and trajectory optimization strategy are verified by the datasets of Petrel-L in the sea trial.

The conclusions are supported by the results, and the provided information is relevant for the knowledge field. Nevertheless, some issues should be addressed before this manuscript could be considered for publication.

1) Full experimental details must be provided so that the results can be reproduced.

2) Detailed implementation information should be provided (hardware, software, configuration, settings).

3) The experimental section, present a large amount of information, it is recommended to summarize the results and provide insight on the advantages and limitation of the proposed methodology, and advice on how to implement it for a specific application (discussing the minimum resources required implementation).

4) The Conclusion section is superficial, should include quantitative results, advantages and disadvantages, limitation and recommendation for real implementations, and future work should be extended.

Round 2

Reviewer 2 Report

The authors addressed the recommendations, the manuscript has been sufficiently improved and could be considered for publication.